# Investigation of Welds and Heat Affected Zones in Weld Surfacing Steel Plates Taking into Account the Bead Sequence

**DOI:** 10.3390/ma13245666

**Published:** 2020-12-11

**Authors:** Miloš Mičian, Jerzy Winczek, Marek Gucwa, Radoslav Koňár, Miloslav Málek, Przemysław Postawa

**Affiliations:** 1Department of Technological Engineering, Faculty of Mechanical Engineering, University of Žilina, 010 26 Žilina, Slovakia; radoslav.konar@fstroj.uniza.sk (R.K.); miloslav.malek@fstroj.uniza.sk (M.M.); 2Department of Technology and Automation, Faculty of Mechanical Engineering and Computer Sciences, Czestochowa University of Technology, 42-201 Czestochowa, Poland; mgucwa@spaw.pcz.pl (M.G.); postawa@ipp.pcz.pl (P.P.)

**Keywords:** weld surfacing, welding sequence, weld geometry, structure analysis, phase shares, hardness distribution

## Abstract

In this paper, the experimental investigation results of the bead sequence input on geometry, structure, and hardness of surfaced layers after multi-pass weld surfacing are analyzed. Three S355 steel plates surfaced by GMAW (Gas Metal Arc Welding) were tested with three different combinations of six beads. The geometric, structural, and hardness analysis was carried out in the cross-section of the plates in the middle of the welded layers. The dimensions of padded layers, fusion and heat-affected zone, as well as the individual padded weld were evaluated. On the basis of metallographic samples, qualitative and quantitative structure analysis was performed. Hardness measurements in surfacing welds and heat-affected zones in the tested cross-sections of the surfacing layers were carried out. A comparative analysis of structure and hardness, taking into account the thermal implications of the bead sequence, allowed for the formulation of conclusions. Comparative studies have shown differences in properties between heat-affected zones (HAZ) for individual surfacing sequences. These differences were mainly in the dimensions of the surfacing layers, the share of structural components, as well as the uniformity of hardness distributions. Finally, the most favorable sequence in terms of structure and hardness distribution, maximum hardness, and range of hardness has been indicated.

## 1. Introduction

Multi-pass welding (joining, rebuilding, surfacing) is widely used in industrial practice and is of interest to many researchers. The thermo-mechanical and metallurgical phenomena occurring during the multi-pass welding process are complicated and coupled, and their effects are still the subject of research and analysis for scientists.

Numerical methods, mainly FEM (Finite Element Method), are most often used in multi-pass welding analysis. Relatively few works concern the results of experimental research. McDonald et al. [1] measured residual stresses in repaired low ferritic CrMoV steel plates using X-ray diffraction techniques and the deep hole drilling method. Vemanaboina et al. [2] evaluated residual stresses in multi-pass dissimilar butt joints using X-ray diffraction.

An experimental analysis of surfaced layers, mainly in the structural aspect as well as corrosion and tribological resistance, was performed in [3,4,5,6,7,8,9,10,11]. The results of metallographic analysis were presented in [3,4,5,6,7,8]. In work [9] displacements (strains) were measured, and in [10] non-destructive tests of welded joints were performed. The authors [11] proposed optimization of multi-pass welding parameters using the Taguchi method. Corrosion resistance was analyzed in [8]. The influence of bead sequences on the properties of welds and heat-affected zones (HAZ) was not discussed in these works. Romek et al. [12] investigated the impact of padding weld shape on their abrasive wear.

The bead sequence has a great influence on the temperature distribution, resulting in welding distortions and residual stresses, as well as changes in the structural and properties of the material after welding. In their analysis of the influence of welding sequences on thermomechanical states, the researchers' interests focus mainly on predicting the displacements (deformations) and the residual stresses after welding. This scope is dominated by research and optimization of the bead sequence when making welded joints, dividing the weld path into multiple seams and taking into account different directions of their execution [13,14,15,16,17,18,19,20,21,22,23,24,25,26,27,28,29,30]. In the works [13,14,15,16,17,18] only the results of numerical analysis were presented for various sequences of beads in simulated states of strain (deformation) and stress in welded joints. In some works, the results of numerical simulations were compared with the experimental results published by other researchers [19,20]. In other studies, the verification of the calculation results was carried out based on experimental investigations. In the works [21,22,23], the verification of the simulated thermal cycles was carried out with thermocouple measurements. Deng [24] used the electric strain gauge method to measure strains in longitudinal and transverse directions. The values of calculated residual stresses using the X-Ray method were verified in [22,25,26]. Displacements and distortions were measured in [27,28,29,30].

Jiang et al. [31] investigated the effect of the number of beads on the quality of a joint in stainless steel clad plates. The results of numerical simulations were compared with experimental data presented in the literature. They showed that with increasing the number of welding layers, the residual stresses are reduced.

There are relatively few works containing the numerical or experimental analysis of the influence of the surfacing bead sequence on the dimensions of heat-affected zones, structure, mechanical, or other properties [32,33].

Kohandehghan and Serajzadeh [32] conducted an analysis of the imposing subsequent welds influence on the development (change in value) of longitudinal and transverse stresses in AA5251 aluminum alloy plates with a thickness of 2 mm. During the tests, only one (serial) surfacing sequence was used. In addition, aluminum alloys do not undergo phase changes (structural changes), so they intensely affect the final structure of the material, mechanical properties, as well as the state of internal stress after welding.

Tomków et al. [33] analyzed the impact of the surfacing sequence in underwater welding on cracks tendency. The hardness structure tests of two hardfacing layers made with different sequence of five stitches were carried out. The first sample was made with serial welds, the second with alternating beads to the left and right of the center of the layer (first bead). As a result of the research, it was found that the susceptibility of steel to cold cracking can be reduced by changing the sequence of the beads during the process.

According to the literature review, the current state of knowledge on the input of the surfacing bead sequence on the geometric parameters of the surfacing layers, their structure and hardness are insufficient. The influence of the weld surfacing sequence on material properties (mainly in weld and HAZ) requires further study considering a larger number of surfacing sequence combinations as well as a larger number of beads. The outermost beads of the surfacing layer, due to the cooling conditions other than for the inner beads of the layer, can differ both structural and mechanical (functional) properties.

In this paper, the influence of three different combinations of six-bead sequences on the geometrical, structural, and hardness analysis of heat-affected zones in three S355 steel plates surfaced by GMAW (Gas Metal Arc Welding) is analyzed. The materials used and research methods were described. Comparative tests were carried out on metallographic specimens (macro and micro) made in half the length of the surfaced layers. The macro samples made it possible to measure the geometric parameters of the surfacing layers (width, surface area, reinforcement height), as well as HAZ. Then, a structural analysis was carried out determining the type of structure. Geometric and structural analysis was performed using a microscope equipped with specialized software. Based on the size and number of grains, the shares of structural components were estimated. The last stage of the experimental tests was the hardness measurements. The article ends with conclusions from the analysis of the results of the experiments performed and the directions of further research.

In welding practice, the choice of surfacing bead sequences is often made on the basis of experience and intuition. Although tests of multi-pass welded joints are conducted [1,2,3,4,5,6,7,8,9,10,11,12,13,14,15,16,17,18,19,20,21,22,23,24,25,26,27,28,29,30], detailed information on the effects of using certain sequences of surfacing beads is very scarce [31,32,33]. In welding handbooks (e.g. [34]) there are multi-pass surfacing schemes for selected machine parts without justifying such a solution. The article deals with this difficult subject by analyzing the influence of the sequence of surfacing beads on the geometrical dimensions of the surfacing layer, structure, and hardness distributions. Based on the analysis of the considered order and place of weld application, the most favorable sequence in terms of structure and hardness distribution, maximum hardness, and range of hardness has been indicated. The results of the research allowed for the designation of further investigation directions.

## 2. Materials and Methods

The investigation was carried out on three S355 steel plates welded with different sequences of six beads. Square plates with a side length of 30 cm and a thickness of 30 mm were surfaced using the Robot CLOOS QRC 350 (CLOOS-Polska, Świdnica, Poland) and the Qineo PULSE 450 (CLOOS, Haiger, Germany) welding source.

The six beads imply 720 combinations of their execution (without taking into account the change of direction—the electrode velocity vector). Therefore, the authors had to select the number of samples, their size, and the technological parameters. The selection of the sequence was made taking into account the increase in the inter-pass times (temperatures) in the B and C plates, based on our own experience and the surfacing sequences adopted, among others in [35], and not to repeat the alternative sequence adopted at work [33].

The size of the welded plates (as well as the number and size of beads) was determined by the dimensions of the samples for further tests, depending on the capabilities of the test equipment.

Before proceeding to a substantial part of the experiment, tests were performed to determine the parameters for a single weld dimensions: a width of 2 cm and a height of approximately 3 mm. Such dimensions of a single padding weld allowed to make a layer of padding welds (taking into account the technological overlap) up to 10 cm wide. The boundary dimensions of the welded zone (in the central part of the plate) resulted from the assumptions of the condition of no heat exchange with the environment through the bottom and side walls of the welded plates.

The additional material was selected so that its chemical composition differed slightly from the chemical composition of the deposited material. The technological parameters were established on the basis of single-pass welding tests, also using the software of the Qineo PULSE 450 welding source.

Each surfacing sequence was performed once. No attempts were made repeatedly because the purpose of the experience was the comparative analysis of qualitative data, not quantitative. Statistical processing of the results was not planned.

All three GMAW welding tests were carried out while maintaining the same welding parameters: voltage 28.3 V, current 290 A, welding speed 0.4 m/min, torch tilt angle 90°, and CTDW (Contact Tip to Work Distance) 15 mm. The welding was made using a Lincoln Ultra MAG G3Si1 (Lincoln Electric Europe, Nijmegen, The Netherlands) wire with a diameter of 1.2 mm, a wire feeding speed of 9.5 m/min, and in MIX 21 (82% Argon/18% CO_2_) shielding gas with a flow rate of 12 L/min. Twenty centimeter-long welds were applied in the central parts of the plates, and the order of individual beads for individual sequences A, B, and C is shown in Figure 1. The inter-operative time associated with the movement of the robot arm to the beginning of the next bead was 3 s.

From the weld-surfaced plates, for metallographic examination and hardness measurements, material strips about 25 mm thick were cut perpendicular to the surfacing direction. Samples were taken at mid-length.

Macroscopic examination of cross-sections of the deposited layers were performed on a Zeiss LSM 700 confocal optical microscope. Metallographic (structural) analysis in the areas of welds and heat-affected zones as well as native material was carried out using Olympus GX51 (Olympus Europe, Hamburg, Germany) metallographic microscope with Olympus Stream Essentials software.

Hardness measurements of individual heat-affected zones in the cross-sections of surfacing welds were made with an automatic hardness tester ZHVµ Micro Vickers hardness device from the company Zwick Roell (Ulm, Germany) in accordance with EN ISO standards [36,37].

## 3. Results and Discussion

### 3.1. Macroscopic Examination

In Figure 2, the macro-panoramas of the specimens in the central cross-section of the plates have been presented.

The measured geometrical values of the welds for individual bead sequences are summarized in Table 1, Table 2 and Table 3. The numbers without asterisks indicate the order of the welds in the layers presented in Figure 2, whereas the numbers with asterisks (in half-round brackets) indicate the order of the bead sequences.

Figure 3, Figure 4 and Figure 5 compare reinforcement height, penetration, and HAZ depths of individual beads in samples A, B, and C numbered in their order in the surfaced layer shown in Figure 2.

The values of reinforcement height and depth of HAZ of beads both between samples and in samples show great variation, with the largest differences in depth of HAZ observed in sample A. Relatively small differences occur in depth of penetrations. Figure 6, Figure 7 and Figure 8 compare reinforcement height, penetration depth, and HAZ depth for samples A, B, and C in the order of beads in accordance with welding sequences.

Although the measured values are varied (also within the samples), an increase in the geometrical parameters can be observed along with the subsequent beads, regardless of their sequence. Especially in sample A, an increase in the depth of penetration is noticeable with subsequent weld beads. The welding thermal cycle of the surfacing caused a successive heating of subsequent areas before laying the next beads.

The geometrical analysis of single beads is of general informative nature about the height of the overlay of the surfacing layer and its fusion depth (HAZ depth). It is obvious that the height of the reinforcement, as well as the fusion depth of the welds depends on the presence of adjacent beads and the technological weld overlay, which is approximately 50% of the bead width. An important piece of information is the smallest reinforcement height among the seams of the applied layer, which may determine the conditions of possible machining (e.g., in the case of rebuilding). In the case of hardfacing, these dimensions are of secondary importance. More useful information is the geometric dimensions of the padding layer, such as its width (Table 1, Table 2 and Table 3), reinforcement, penetration, HAZ, and total weld padding areas.

In Figure 9, the areas of individual zones in the weld layers: reinforcement, penetration, HAZ, and total weld padding areas are presented. The sizes of these areas are the largest for sample B (sequence 5-3-1-2-4-6) and the lowest for sample A (sequence 1-2-3-4-5-6. However, it is difficult to say what the reason is. In the case of sample C (sequence 1-4-2-5-3-6), intermediate values (between samples A and B) were recorded for all measured areas.

### 3.2. Microscopic Investigation

The parent material was S355 steel plates with a ferritic–pearlitic structure in a banded system after rolling (Figure 10). Structural analysis in the areas of welds and heat-affected zones as well as native material was performed using Olympus GX51 metallographic microscope with Stream software. The structure of padding welds was tested in their central part; Figure 11 indicates places in HAZ where structural components are assessed. Shares of structures in indicated places of the surfaced welds for individual plates are listed in Table 4 (sample A), Table 5 (sample B), and Table 6 (sample C).

Figure 12 shows the typical changes in structure after welding. There is a small but visible amount of the martensite close to the fusion line in front of the first bead and in the axis of the bead. The structure in HAZ is not homogenous due to the effect of the welding cycle and some chemical segregation of the base material, which is visible in Figure 10 (lamellar structure after plastic deformation and Mn segregation). Farther from the fusion line structure is a mix of bainite and perlite.

In sample A, in addition to the effect of increased penetration depth, you can see changes in the HAZ width and changes in its structure. As a result, near the fusion line for the fifth and sixth bead you can observe elements of the microstructure similar in appearance to the Widmanstätten ferrite, which testify to a large overheating of the structure (Figure 13a).

Additionally, the structure between the last beads is gradually degraded, because in addition to the mixture of martensite and bainite, Widmanstätten ferrite grains appear (Figure 13b), which can potentially lead to weakening of the structure in this area.

In the HAZ structure, the bainitic structure dominates, with a small amount of martensite between successive beads. Martensite can be mainly seen near the fusion line and its quantity decreases significantly with increasing distance from the fusion line. Depending on the observation site, a change in bainite morphology can be seen.

In the central areas of sample B, which were first surfaced, a large amount of lower bainite can be seen (Figure 14a), whereas in the HAZ between the outer stitches made at the end, lower bainite occurs in a smaller amount (Figure 14b).

In Figure 15a, the structure of the HAZ near the fusion line in front of stitch 1 for sample C is shown. The structure in this area consists mainly of martensite, which accounts for about 70% of the structure in the analyzed area. This is also confirmed by the hardness test results (please compare with Figure 19), where the highest hardness value was measured for this area.

The structure in Figure 15b is different in the quantity of fraction (Table 6) compared with the structure shown in Figure 15a. Because this area (in the axis of the bead number one) has the longest cooling time, the structure consists mainly of bainite with a small amount of the martensite. Farther from this area, the structure consists of mixed bainite, perlite, and ferrite. In the axis of bead number five (Figure 16), there is mainly fine bainitic structure close to the fusion line. This bead was done in the middle of the plate and was the penultimate bead, so the heat accumulated in the plate increased the time of cooling and prevented formation of martensite.

The changes in structure of investigated plates are mainly in the quantity of the phases in the characteristic areas of the HAZ. Martensite structure is obtained almost in whole researched regions after the cladding process. The differences in the quantity of the martensite structure is because of the level of the heat input, heat dissipation, and sequence of the cladding. For the first external beads in each case (plates A, B, and C), there are noticeable amounts of the martensite in front of the beads. In the middle of the plates, especially for plate C, the amount of the martensite decreases because of the accumulated heat and extended cooling time. The martensite was observed close to the fusion line and there were some areas very rich in this structure. This can be explained by the segregation of the C and Mn in the structure of the base material (the texture after plastic deformation). The short heating time and fast cooling time limited the diffusion of the elements and created conditions for the formation of the martensite in certain areas of the HAZ.

### 3.3. Hardness Measurement Results

The hardness HV0.5 was measured in a line 2 mm away from the surface of the beads. It passes from the basic material through the individual welds and their HAZ. The measurement was performed on an automatic hardness tester ZHVµ Micro Vickers hardness device from the company Zwick Roell with automatic reading of the size of the indentation. The distance between indentations was 250 µm.

The sequence of welding beads has a significant effect on the hardness distribution in the measured line. From the course of hardness, the mutual influence of individual beads is visible. The hardness of HAZ is reduced by the effect of annealing with another weld bead. Figure 17, Figure 18 and Figure 19 show macroscopic images of the individual sequences with the measurement line and a graphical representation of the hardness distribution.

External weld beads reach higher values of hardness in HAZ than internal ones. This effect was observed for all sequences. In each of them a high content of martensite is observed near the fusion line, which results from the rapid heat dissipation during the laying of the first weld beads.

In the case of sequence A, the hardness of the weld bead and its HAZ decrease with the consecutive application of further beads. Heat accumulates in the base material and increases inter-pass temperature. This increases the time t_8/5_, leading to the transformation of austenite into softer microstructures. The maximum individual hardness value is 391 HV (in the area of first HAZ), the minimum is 201 HV (in the area of HAZ between the fifth and sixth welding beads).

In the case of sequence B, the welds are deposited from the center alternately on one and the other side. The inner beads are affected by others, which leads to the annealing of microstructure. The hardness of the welding beads is approximately the same (about 240 HV), and the HAZ reach even higher hardness values. External welds reach significantly higher values in HAZ (with maximum individual value of 350 HV) because they were made last. The effect of increased hardness in these places was not affected by the rising inter-pass temperature.

In the case of sequence C, an alternating application of beads is utilized. Beads four and five have a significant annealing effect and the hardness in this section is relatively balanced. The maximum hardness value is 345 HV (in HAZ of first bead). In terms of hardness distribution in the measured line, maximum hardness, and range of hardness, sequence C has the most suitable properties. The alternating (1-4-2-5-3-6) sequence of depositing weld seams had a positive effect on the hardness distribution in the joint. The hardness values had a low variance, the cooling time t_8/5_ was extended. In the bead axis, the dominant structure is bainite. This applies not only to this case, but for almost every bead in the tested samples.

The hardness values from the measured lines were used to determine the average hardness of individual HAZ and welding beads. Figure 20a shows the average hardness values for individual welding beads for all welding sequences. In addition, Figure 20b shows the average hardness values for the individual heat-affected zones and all welding sequences. These values also support the microscopic determination of the volume percentage of structural phases in the individual zones evaluated in Section 3.2.

## 4. Conclusions

The experimental investigation results of the bead sequence on characteristic geometric dimensions, structure, and hardness of welds and heat-affected zones allow one to formulate the following conclusions:The serial sequence of the beads results in an increase in the depth of fusion along with the laying of subsequent surfacing beads. This is probably due to the length of the welds (time of making a single run), leading to a very short inter-pass time (increasing inter-pass temperature in the consecutive beads) or not obtaining a quasi-stationary temperature distribution in the padding welds. Determining the inter-pass temperature is important not only for the structure and hardness of the material after surfacing but also for the depth of penetration (HAZ).The calculated reinforcement, penetration, HAZ, and total weld padding areas are the largest for sample B (sequence 5-3-1-2-4-6) and the lowest for sample A (sequence 1-2-3-4-5-6 ). For sample C (sequence 1-4-2-5-3-6), intermediate values (between samples A and B) were recorded for all measured fields.Analysis of the geometrical dimensions of the weld and HAZ does not allow to formulate unambiguous conclusions regarding the influence of the welding sequence on the depth of penetration and depth of HAZ for individual beads.The greatest width, reinforcement, penetration, and total weld area in the cross-sections of the padded layer were obtained from sample B, and the lowest from sample A.The dominant structure in HAZ is bainite and does not depend on the bead sequence in contrast to the martensitic structure, which is most visible in the HAZ before the first bead of the weld and in the HAZ areas between successive overlapping beads.The most favorable structure was observed in sample C, where the alternating (1-4-2-5-3-6) sequence of depositing bead welds had a positive effect on the hardness distribution (maximum values and range) in the joint by extending the cooling time, creating a uniform temperature distribution and limiting the steel's tendency to hardening.

The achieved research results, their analysis, and their conclusions do not give an unequivocal answer to the question of which sequence of beads may be the most advantageous. Therefore, experimental analysis of the influence of the bead sequence on the properties of surfacing welds and HAZ should be the subject of further research and analysis, as it may be particularly important in the case of rebuilding and hardfacing layers. In the case of hardfacing with the use of appropriate electrode wires, the aim should be to obtain a regular distribution of high hardness (avoiding tempered zones). The use of additional research methods to determine the effect of the surfacing sequence on the type and magnitude of stresses can be very helpful in determining areas with a potentially higher probability of crack formation (rebuilding) and areas with reduced wear resistance (hardfacing). An important aspect would also be to determine the impact of the surfacing sequence on the type and size of welding distortions.

## Figures and Tables

**Figure 1 materials-13-05666-f001:**
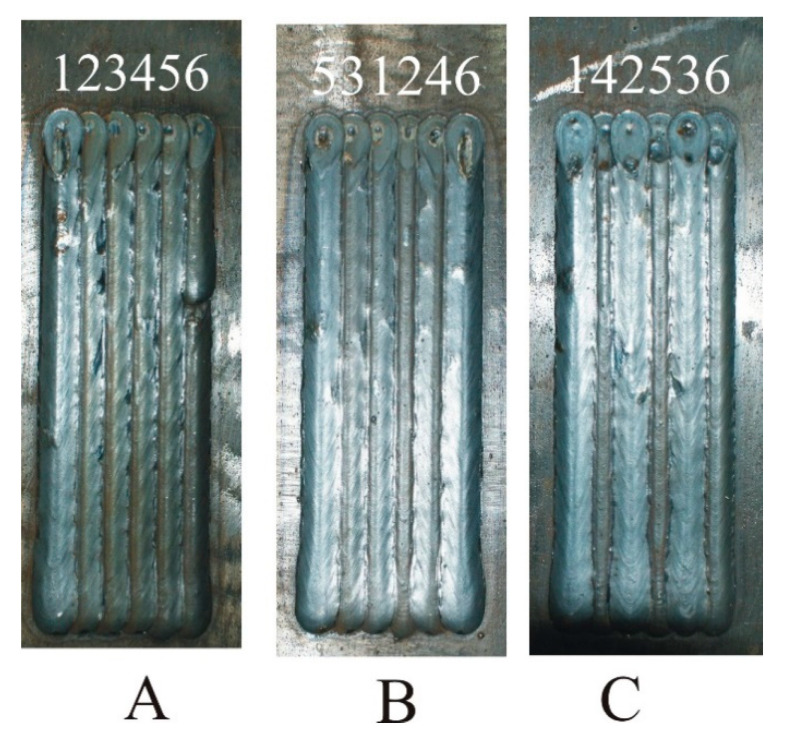
Photographs of plates with padded welds marked by a sequence of beads.

**Figure 2 materials-13-05666-f002:**
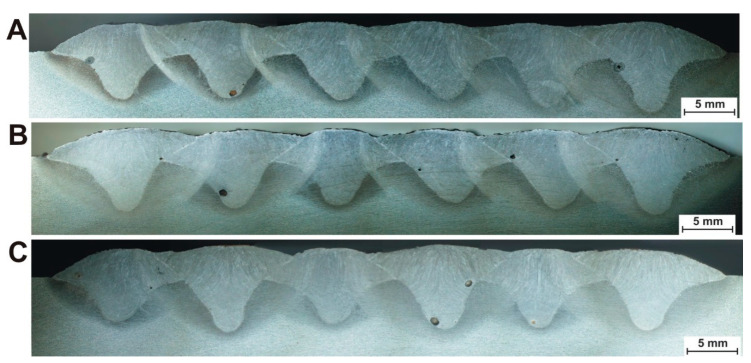
Macro-panoramas of the metallographic specimens in the middle cross-section of the plates.

**Figure 3 materials-13-05666-f003:**
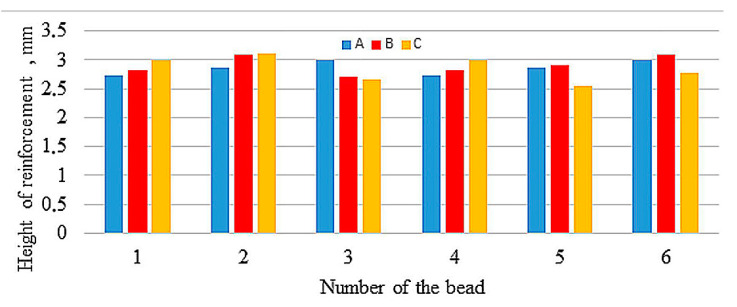
Reinforcement height of particular beads.

**Figure 4 materials-13-05666-f004:**
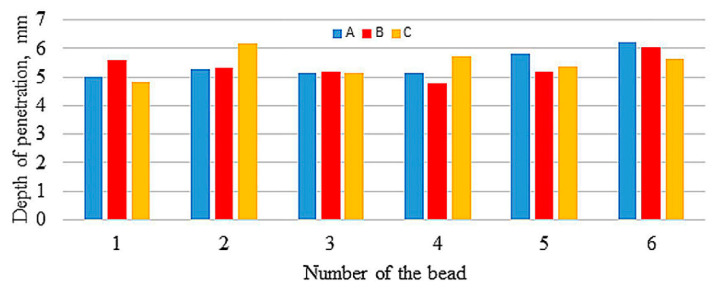
Penetration depth of particular beads.

**Figure 5 materials-13-05666-f005:**
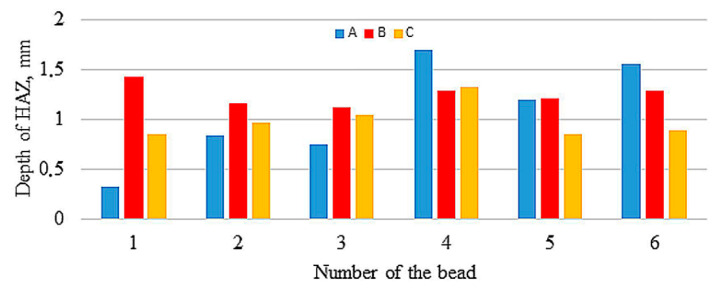
Heat-Affected Zone (HAZ) depth of particular beads.

**Figure 6 materials-13-05666-f006:**
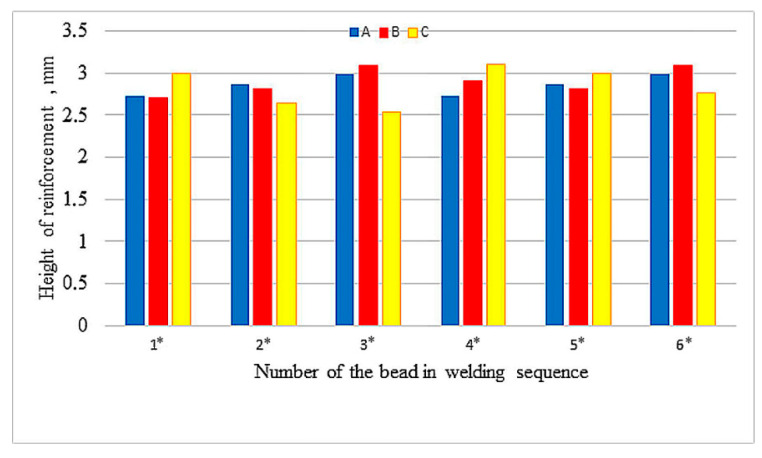
Height of the reinforcement in welding sequences.

**Figure 7 materials-13-05666-f007:**
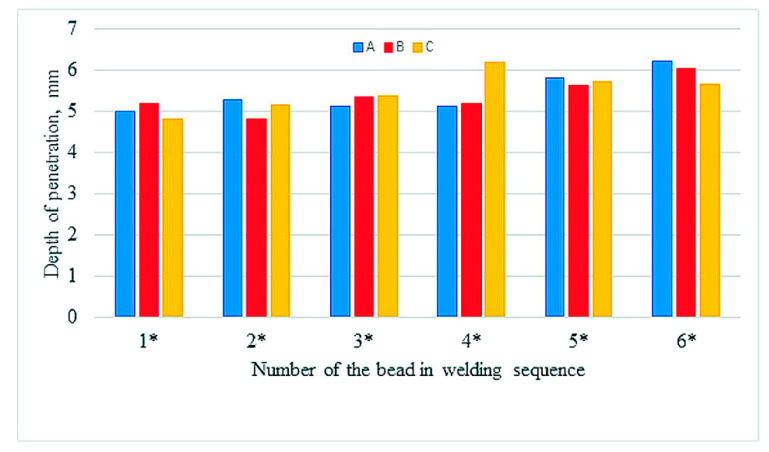
Depth of the penetration in welding sequences.

**Figure 8 materials-13-05666-f008:**
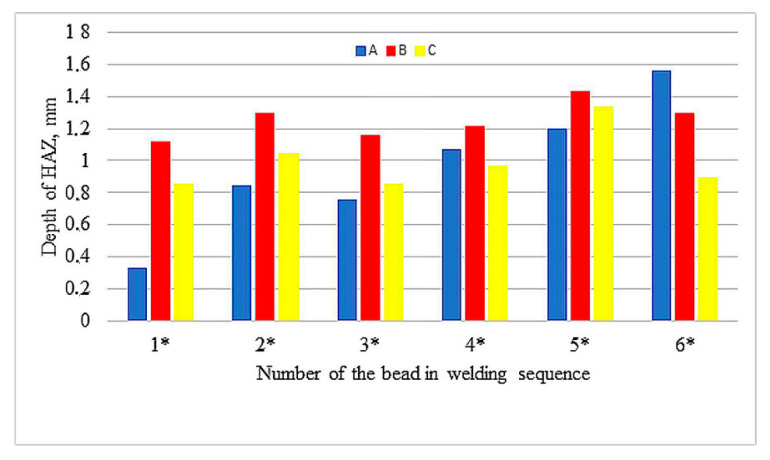
Depth of the HAZ in welding sequences.

**Figure 9 materials-13-05666-f009:**
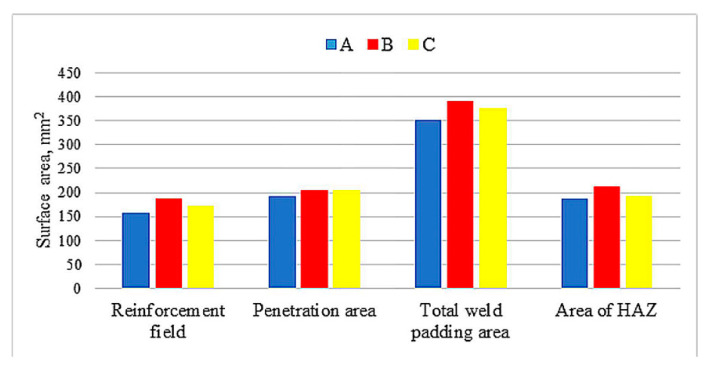
Surface areas of individual zones in the weld layer cross-sections.

**Figure 10 materials-13-05666-f010:**
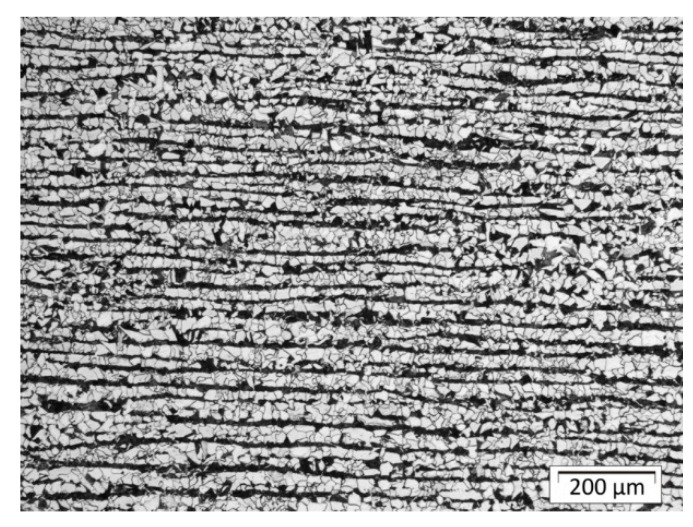
Native material: ferritic–pearlitic in a banded system after rolling.

**Figure 11 materials-13-05666-f011:**
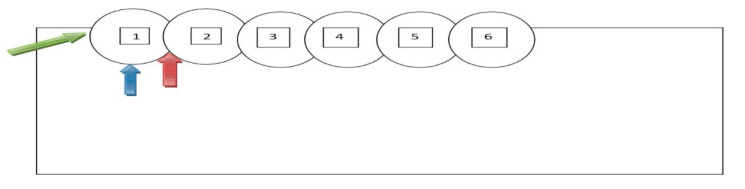
Places in HAZ where structural components were assessed.

**Figure 12 materials-13-05666-f012:**
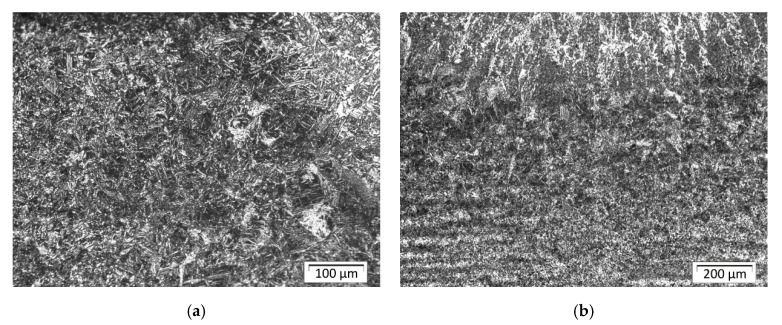
Sample A: (**a**) HAZ in front of bead number one; (**b**) fusion line and HAZ in axis of bead number one.

**Figure 13 materials-13-05666-f013:**
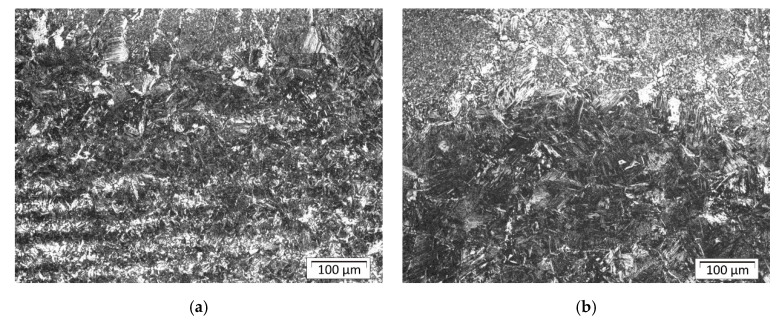
Sample A: (**a**) the structure of the HAZ in the axis of bead number six; (**b**) the structure of the HAZ between the fifth and sixth bead.

**Figure 14 materials-13-05666-f014:**
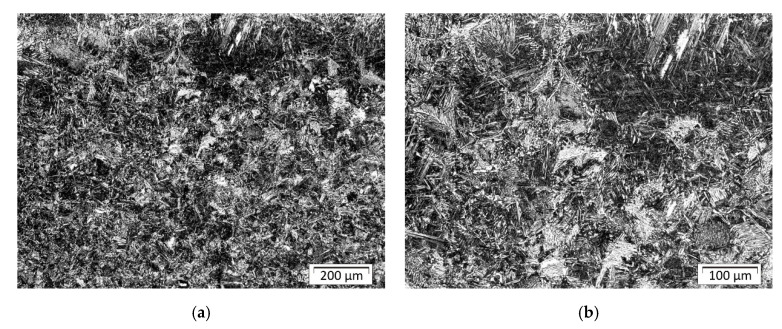
Sample B: (**a**) the structure of the HAZ between the first and second bead; (**b**) the structure of the HAZ between fifth and third bead.

**Figure 15 materials-13-05666-f015:**
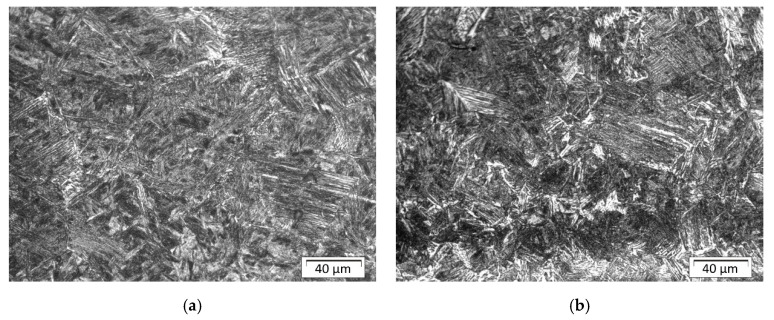
Sample C: (**a**) the structure of the HAZ in front of bead number one; (**b**) the structure of the HAZ close to the fusion line in the axis of bead number one.

**Figure 16 materials-13-05666-f016:**
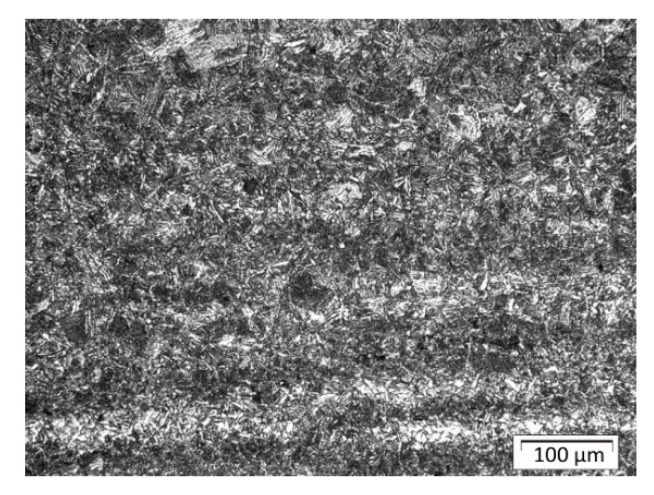
Sample C: the structure of the HAZ in the axis of bead number five.

**Figure 17 materials-13-05666-f017:**
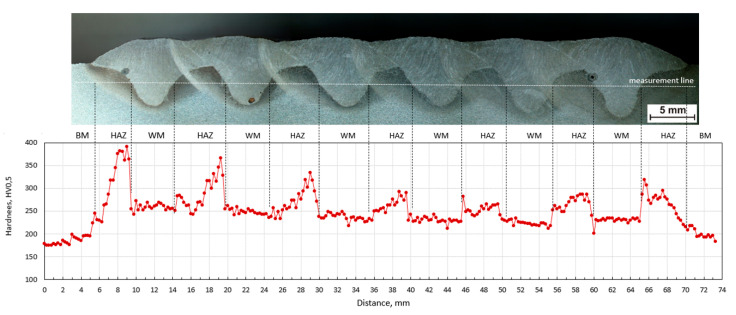
Hardness HV_0.5_ distribution in plate A; BM—base material, W—weld metal, HAZ—heat-affected zone.

**Figure 18 materials-13-05666-f018:**
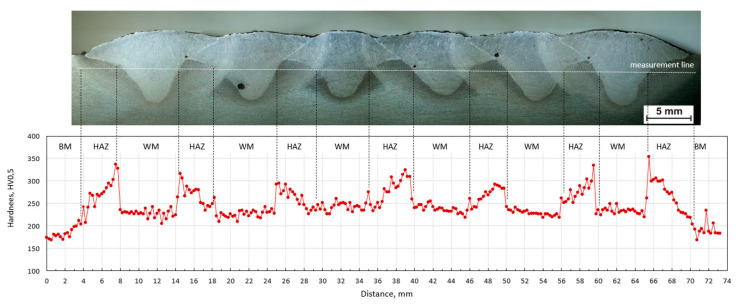
Hardness HV_0.5_ distribution in plate B; BM—base material, WM—weld metal, HAZ—heat-affected zone.

**Figure 19 materials-13-05666-f019:**
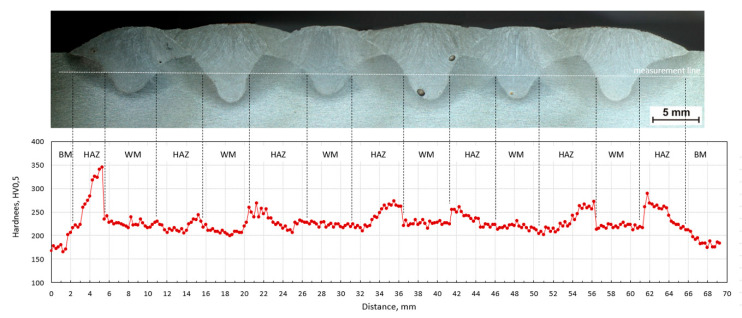
Hardness HV_0.5_ distribution in plate C; BM—base material, WM—weld metal, HAZ—heat-affected zone.

**Figure 20 materials-13-05666-f020:**
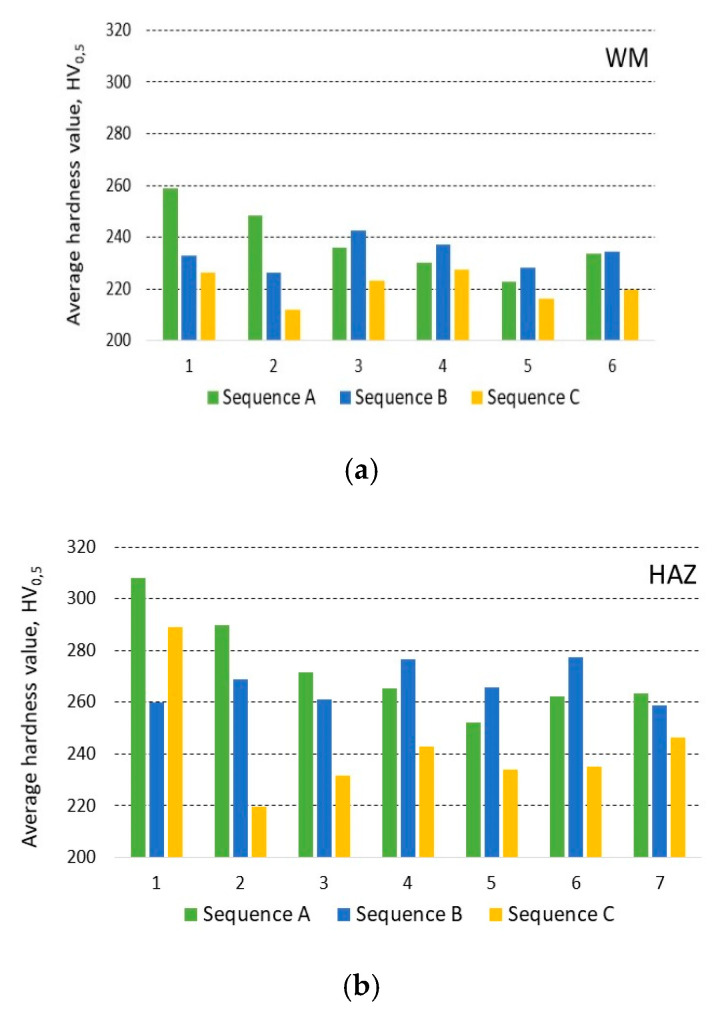
Average hardness value HV_0.5_ of (**a**) weld metal (WM) and (**b**) of heat-affected zone (HAZ), measured on cross-sections of plates from left to right.

**Table 1 materials-13-05666-t001:** The geometrical characteristics of the deposition zone in plate A.

Width of padded area, mm	69.11					
Number of the weld	1(1*)	2(2*)	3(3*)	4(4*)	5(5*)	6(6*)
Depth of penetration (fusion), mm	5	5.27	5.13	5.13	5.8	6.21
eight of reinforcement, mm	2.72	2.86	2.99	2.72	2.86	2.99
Depth of HAZ, mm	0.33	0.844	0.756	1.07	1.2	1.56
Reinforcement field, mm^2^	157.36					
Penetration area, mm^2^	192.03					
Total weld padding area, mm^2^	349.39					
Area of HAZ, mm^2^	185.81					

**Table 2 materials-13-05666-t002:** The geometrical characteristics of the deposition zone in plate B.

Width of padded area, mm	72.13					
Number of the weld	1(5*)	2(3*)	3(1*)	4(2*)	5(4*)	6(6*)
Depth of penetration (fusion), mm	5.61	5.34	5.2	4.8	5.2	6.02
Height of reinforcement, mm	2.81	3.08	2.71	2.81	2.9	3.08
Depth of HAZ, mm	1.43	1.16	1.12	1.29	1.21	1.29
Reinforcement field, mm^2^	186					
Penetration area, mm^2^	204.88					
Total weld padding area, mm^2^	390.88					
Area of HAZ, mm^2^	211.36					

**Table 3 materials-13-05666-t003:** The geometrical characteristics of the deposition zone in plate C.

Width of padded area, mm	71.03					
Number of the weld	1(1*)	2(4*)	3(2*)	4(5*)	5(3*)	6(6*)
Depth of penetration (fusion), mm	4.81	6.18	5.15	5.72	5.38	5.65
Height of reinforcement, mm	2.99	3.11	2.65	2.99	2.54	2.77
Depth of HAZ, mm	0.852	0.963	1.04	1.33	0.852	0.889
Reinforcement field, mm^2^	172.74					
Penetration area, mm^2^	202.99					
Total weld padding area, mm^2^	375.73					
Area of HAZ, mm^2^	190.64					

**Table 4 materials-13-05666-t004:** Shares of structures in the surfaced welds on plate A.

Type of Structure	Structure Share%								
	M	B	F + P	M	B	F + P	M	B	F + P
Number of bead	1			2			3		
In front of bead	20	40	40						
In the bead axis	5	50	45	0	60	40	0	60	40
Between current and next bead	30	40	30	20	50	30	20	40	40
Number of bead	4			5			6		
In the bead axis	10	50	40	0	40	60	10	50	40
Between current and next bead	10	40	50	20	40	40			
Behind bead							10	40	50

**Table 5 materials-13-05666-t005:** Shares of structures in the surfaced welds on plate B.

Type of Structure	Structure Share%								
	M	B	F + P	M	B	F + P	M	B	F + P
Number of bead	5			3			1		
In front of bead	20	60	20						
In the bead axis	0	40	60	0	40	60	0	70	30
Between current and next bead	30	40	30	5	60	35	10	60	30
Number of bead	2			4			6		
In the bead axis	20	40	40	0	60	40	0	50	50
Between current and next bead	10	40	50	10	50	40			
Behind bead							5	45	50

**Table 6 materials-13-05666-t006:** Shares of structures in the surfaced welds on plate C.

Type of Structure	Structure Share%								
	M	B	F + P	M	B	F + P	M	B	F + P
Number of bead	1			4			2		
In front of bead	30	50	20						
In the bead axis	10	40	50	0	40	60	0	70	30
Between current and next bead	10	40	50	0	70	30	10	60	30
Number of bead	5			3			6		
In the bead axis	0	40	60	0	40	50	0	50	50
Between current and next bead	10	50	40	5	55	40			
Behind bead							10	40	50

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
