# Peer review of "Investigation of Welds and Heat Affected Zones in Weld Surfacing Steel Plates Taking into Account the Bead Sequence"

_materials, 2020, doi:10.3390/ma13245666_

Round 1

Reviewer 1 Report

The authors investigated the effect of three bead sequences on geometry, structure and hardness of surfaced layers of S355 steel plates after multi-pass weld surfacing through Gas Metal Arc Welding. However, they showed a similar trend for all the scenarios and therefore not allowing finding the best sequence.

In general, the manuscript is well written and the results are clearly presented. While, some concerns arise from the experimental approach adopted and should be further explained.

Major comments:

Lines 25-26: the last part of the abstract describes the approach but not the results. Please, add some information about the obtained results.

Line 27: the last two keywords are too general, please rewrite them being more specific for the case study.

Lines 87-90: in the last part of the introduction, where authors describe the work carried out, the methods and most important results are missing. Please, add some more information about those aspects.

Lines 95-97: the authors do not justify the adoption of such parameters. Do they carried out some preliminary investigation? Please, clarify. Moreover, how many replications did the authors perform? Each sequence was replicated just once? Why did they choose such sequences?

Minor comments:

Figures 12 to 15: please, add a greater scale bar to make it more easily readable.

Please, make the conclusions more compact. In the following some suggestions:

Line 294: this conlusion (the 4th bullet) is the same of the second.

Line 311: the last conclusion (last bullet) can be merged with the 5th.

Lines 303-310: please, merge this conclusions together (6th to 9th bullet).

Author Response

The authors would like to thank to reviewer for his valuable comments and suggestions which have been subsequently used to improve the manuscript. All the comments and suggestions have been carefully addressed in the revised manuscript. The changes made in manuscript are highlighted in yellow colour. The itemized response to the comments is provided in the following.

Comments and Suggestions for Authors

The authors investigated the effect of three bead sequences on geometry, structure and hardness of surfaced layers of S355 steel plates after multi-pass weld surfacing through Gas Metal Arc Welding. However, they showed a similar trend for all the scenarios and therefore not allowing finding the best sequence.

In general, the manuscript is well written and the results are clearly presented. While, some concerns arise from the experimental approach adopted and should be further explained.

Major comments:

Comment: Lines 25-26: the last part of the abstract describes the approach but not the results. Please, add some information about the obtained results.

Response: The abstract was supplemented with information about the obtained results.

Comment: Line 27: the last two keywords are too general, please rewrite them being more specific for the case study.

Response: The last two keywords have been rewritten.

Comment: Lines 87-90: in the last part of the introduction, where authors describe the work carried out, the methods and most important results are missing. Please, add some more information about those aspects.

Response:  The research description has been supplemented as suggested by the reviewer. In addition, the methods are presented in the Materials and methods chapter.

Comment: Lines 95-97: the authors do not justify the adoption of such parameters. Do they carried out some preliminary investigation? Please, clarify. Moreover, how many replications did the authors perform? Each sequence was replicated just once? Why did they choose such sequences?

Response:  The six beads imply 720 combinations of their execution (without taking into account the change of direction - the electrode velocity vector). Therefore, the authors had to select both the number of samples, their size and technological parameters. The selection of the sequence was made taking into account the increase in the inter-pass times (temperatures) in the B and C plates, based on our own experience and the surfacing sequences adopted, among others, in the work:

  • Winczek J., Modeling of temperature field during multi-pass GMAW surfacing or rebuilding of steel elements taking into account the heat of the deposit metal, Applied Sciences, 2017, 7(1), 6; doi:10.3390/app7010006, 1 – 19

and not to repeat the alternative sequence adopted at work:

  • Tomków, J.; Fydrych, D.; Rogalski, G. Role of bead sequence in underwater welding. Materials 2019, 12, 3372; DOI:10.3390/ma12203372.

The size of the welded plates (as well as the number and size of beads) was determined by the dimensions of the samples for further tests, depending on the capabilities of the test equipment.

Before proceeding to a substantial part of the experiment was performed a single test beads to determine the parameters for a single weld dimensions: a width of 2 cm and a height of approx. 3 mm. Such dimensions of a single padding weld allowed to make a layer of padding welds (taking into account the technological overlap) up to 10 cm wide. The boundary dimensions of the welded zone (in the central part of the plate) resulted from the assumptions of the condition of no heat exchange with the environment through the bottom and side walls of the welded plates.

The additional material was selected so that its chemical composition differed slightly from the chemical composition of the deposited material. The technological parameters were established on the basis of single-pass welding tests, also using the software of the Qineo PULSE 450 welding source.

Each surfacing sequence was performed once. No attempts were made repeatedly, because the purpose of the experience was the comparative analysis of qualitative, not quantitative. Statistical processing of the results was not planned.

The authors did not include this information in the article because, in their opinion, they are not important for the conducted experiments, the analysis of the research results and the formulated conclusions.

Minor comments:

Comment: Figures 12 to 15: please, add a greater scale bar to make it more easily readable.

Response: The scale in the figures has been increased.

Comment: Please, make the conclusions more compact. In the following some suggestions:

Line 294: this conclusion (the 4th bullet) is the same of the second.

Line 311: the last conclusion (last bullet) can be merged with the 5th.

Lines 303-310: please, merge this conclusions together (6th to 9th bullet).

Response:  Improvements made according to reviewer's suggestions.

Reviewer 2 Report

- Authors should explain the academic contribution of the work developed. Highlighting what is innovative / original about the existing literature.

-Authors should develop the conclusions of the work and refer in more detail to the next steps of the work.

-Authors should explain better the figures 17,18 and 19

- In the Introduction section, authors should describe the structure of the paper.

Author Response

Comment: Authors should explain the academic contribution of the work developed. Highlighting what is innovative / original about the existing literature.

Response:  The reviewer's comment was taken into account. The explain the academic contribution of the work is included at the end of the Introduction chapter.

Comment: Authors should develop the conclusions of the work and refer in more detail to the next steps of the work.

Response:  The reviewer's comment was taken into account. Authors refer in more detail to the next steps of the work at the end of the Conclusion chapter.

Comment: Authors should explain better the figures 17,18 and 19.

Response:  Section 3.3 has been extended to include a more detailed analysis of Figures
17 - 19.

Comment: In the Introduction section, authors should describe the structure of the paper.

Response:  As suggested by the reviewers, the Introduction was supplemented with a description of the scope of the research carried out.

Round 2

Reviewer 1 Report

The authors greatly improved the overall quality of the manuscript by carefully following and accommodating the reviewers' comments/suggestions. Now the experimental approach is more clear thanks to the explanation given in the authors' response document. However, the reviewer suggests to add this information also in the manuscript in Section 2 Materials and Methods to inform the reader how they made the choice of both the operational parameters and the bead sequences.

Author Response

Response to Reviewer #1

The authors would like to thank to reviewer for his valuable comments and suggestions which have been subsequently used to improve the manuscript. All the comments and suggestions have been carefully addressed in the revised manuscript. The changes made in manuscript are highlighted in yellow. The itemized response to the comments is provided in the following.

Comments and Suggestions for Authors

Comment: The authors greatly improved the overall quality of the manuscript by carefully following and accommodating the reviewers' comments/suggestions. Now the experimental approach is more clear thanks to the explanation given in the authors' response document. However, the reviewer suggests to add this information also in the manuscript in Section 2 Materials and Methods to inform the reader how they made the choice of both the operational parameters and the bead sequences.

Response:  The reviewer's suggestion was taken into account and additional information was included in the manuscript in Section 2 Materials and Methods to inform the reader how authors made the choice of both the operational parameters and the bead sequences.
